# Nano-Indentation Properties of Tungsten Carbide-Cobalt Composites as a Function of Tungsten Carbide Crystal Orientation

**DOI:** 10.3390/ma13092137

**Published:** 2020-05-05

**Authors:** Renato Pero, Giovanni Maizza, Roberto Montanari, Takahito Ohmura

**Affiliations:** 1Department of Industrial Engineering, University of Rome “Tor Vergata”, 00133 Rome, Italy; renato.pero@alumni.uniroma2.eu (R.P.); roberto.montanari@uniroma2.it (R.M.); 2Department of Applied Science and Technology, Politecnico di Torino, 10129 Torino, Italy; 3High Strength Materials Group, National Institute for Materials Science (NIMS), 1-2-1 Sengen, Ibaraki Tsukuba 305-0047, Japan; ohmura.takahito@nims.go.jp

**Keywords:** nanoinstrumented indentation test, cermets, metal matrix composites, anisotropy, metallography, atomic force microscopy, indentation properties, tungsten carbide

## Abstract

Tungsten carbide-cobalt (WC-Co) composites are a class of advanced materials that have unique properties, such as wear resistance, hardness, strength, fracture-toughness and both high temperature and chemical stability. It is well known that the local indentation properties (i.e., nano- and micro-hardness) of the single crystal WC particles dispersed in such composite materials are highly anisotropic. In this paper, the nanoindentation response of the WC grains of a compact, full-density, sintered WC-10Co composite material has been investigated as a function of the crystal orientation. Our nanoindentation survey has shown that the nanohardness was distributed according to a bimodal function. This function was post-processed using the unique features of the finite mixture modelling theory. The combination of electron backscattered diffraction (EBSD) and statistical analysis has made it possible to identify the orientation of the WC crystal and the distinct association of the inherent nanoindentation properties, even for a small set (67) of nanoindentations. The proposed approach has proved to be faster than the already existing ones and just as reliable, and it has confirmed the previous findings concerning the relationship between crystal orientation and indentation properties, but with a significant reduction of the experimental data.

## 1. Introduction

Tungsten carbide (WC) bonded with 6 to 20 wt% of cobalt, which is also known as WC-Co composite, cermet, cemented carbide or hard metal, is a family of advanced materials that exhibit unique wear resistance, hardness, strength and both temperature and chemical stability properties combined with a good fracture toughness [1]. WC-Co composites are widely employed in the tool industry for high-speed cutting tools, finishing rolling rings for cold rolling and blades, as well as in the mining industry. Such composites have a soft Co matrix that is intimately bonded to micrometric (or submicrometric) WC particles. The individual WC particles are frequently hexagonal single crystals [2] that possess highly anisotropic mechanical properties [3,4,5]. However, as the WC particles are randomly oriented and uniformly dispersed in the Co matrix, the macroscopic mechanical properties of the composite are isotropic [6]. The mechanical properties of WC-Co composites may retain their original high crystal anisotropy over a nano- and micro-scale range, depending on the local crystal orientation of the WC particles and their surroundings [2,6,7,8,9,10,11,12,13,14,15]. Experiments on WC-Co composites, which have combined nanoindentation tests and angle orientation measurements, based on electron backscattered diffraction (EBSD) analysis, have qualitatively confirmed that the hardness and the indentation modulus of the WC {0001} (basal) plane are larger than that of a WC {1010} (prismatic) plane [2,7,8,9,10,11,12,13,14,15]. Similar results have been found for monolothic WC single crystal counterparts [3,4,5]. Csanádi et al. [2] reported that the hardness of the crystal planes, tilted at an angle of between 0° to 17° with respect to the basal plane, compared well with that of the basal plane. Such planes have been denoted as basal-like planes. Analogous hardness considerations have led to the definition of prismatic-like planes being tilted at an angle of between 37° to 90° [2]. Roa et al. suggested an experimental methodology to establish the relationship between indentation properties and WC crystal orientation by means of a massive nanoindentation test campaign (up to 200,000 tests), using an automated nanoindenter device [15]. Despite the encouraging results, this kind of technique is considered impractical, especially at an industrial scale. Moreover, it is also evident from the literature that the indentation properties of WC-Co composite materials of engineering interest are often in quantitative disagreement. A multitude of factors, including the chemical composition (WC/Co ratio and other additives), morphology, grain size and distribution, manufacturing process (with inherent internal stresses and residual porosity), surface finish, indenter shape, indentation conditions (maximum load and loading rate), indentation size effect and even the procedure used to extract the properties, may influence the mechanical response of the WC-Co composites at any testing scale.

The present work has been aimed at developing a new experimental methodology, which relies on a limited number of nanoindentation tests, to successfully relate the indentation properties of WC grains (dispersed in a WC-10Co composite) to their crystal orientation. This paper is a preliminary study of a wider industrial oriented research that is focused on the mechanical characterisation of solid-state tool-steel/WC-Co welded joints by means of nanoindentation [16]. The nanoindentation property distributions were sampled using a suitable Gaussian mixture model [17]. An efficient algorithm of this model was implemented in the MATLAB® (Mathworks, Natick, MA, USA) software with the Fit Gaussian Mixture Model to Data function [18]. This built-in function permits the desired functional relationship between the indentation properties and the WC crystal orientation to be easily extracted. EBSD, scanning electron microscopy and atomic force microscopy measurements were conducted and the obtained information on the crystal orientation of the WC particles was used to support the statistical analysis.

## 2. Materials and Methods

The sample consisted of a full-density, sintered WC with a 10 wt% Co composite rod (40 mm long and 6 mm in diameter) supplied by Silmax S.p.A. (Lanzo, Italy). The WC particles exhibited a typical, original, faced crystal structure with an average 0.4 μm particle size. The rod samples were first cut longitudinally, across the diameter, into two halves. The opposite base surface of each rod was then milled and ground until they were perfectly flat and parallel to one another, in order to permit nanoindentation. Before indentation, both surfaces were mechanically polished, using SiC papers, from a P80 to a P2400 grit and then a 9 μm diamond grain suspension. Mirror finish surfaces were chemomechanically obtained by using a colloidal silica suspension for 40 min. Finally, the surfaces were rinsed in distilled water and ultrasonically cleaned for 10 min in ethanol.

Prior to indentation, the sample was inspected by means of field emission scanning electron microscopy (FESEM, JSM-7000F, Jeol, Tokyo, Japan), equipped with EBSD and energy dispersive X-ray spectroscopy (EDS, Apollo SDD, EDAX, Foster City, CA, USA) detectors to establish the crystal orientation and to determine the chemical composition, respectively.

Nanoinstrumented indentation tests (nIITs) were performed using a nanoindenter (TI-950 Hysitron Triboindenter, Bruker Co., Billerica, MA, USA) and a Berkovich indenter with a round tip radius of 500 nm. In order to minimise any adverse thermal drift and to permit thermal homogenisation of the overall system, the sample was held in its holder inside the housing for approximately 12 h before the experiment was run. Atomic force microscopy (AFM, TI-950 Hysitron Triboindenter, Bruker Co., Billerica, MA, USA) scanning was employed to identify locations that were sufficiently flat, smooth and large to be indented, and to observe the residual imprints.

In order to allow comparisons to be made with other researches on WC-Co composites [8,10,11,12,14], a maximum test load of 10 mN was selected for the nanoindentation survey. As the authors did not assess larger maximum loads than 10 mN, the absolute absence of any indentation size effect cannot in principle be excluded. However, this influence, if any, is dealt with in the Discussion of the Results section where the attained indentation properties are compared with literature values.

Moreover, it was reported [14] that when the maximum indentation load was lower than 10 mN, the projected area function values measured by means of AFM differed to a great extent from the values estimated through the standard code [19]. We selected a holding time of 10 s and a loading/unloading rate of 50 µN·s^−1^. A set of 75 nanoindentations was performed over the WC region to enable particles with different orientation angles to be sensed.

The piezo actuator attached to the nanoindenter has nominal intrinsic and nonlinear creep responses that are detrimental to the measurements. This detrimental effect was avoided by lagging two subsequent measurements of approximately 3 min. Out of the 75 nanoindentations, only three (not shown) exhibited severe thermal drift (i.e., larger than 0.2 nm s^−1^ or smaller than −0.2 nm·s^−1^) and these were discarded. Furthermore, five nanoindentation curves (not shown) exhibited abnormal contact at the initial loading stage and were thus also discarded. The remaining 67 nanoindentation tests were post-processed.

It can be anticipated that the present EBSD analysis conducted over nonindented WC particles elucidated that triangular- and rectangular-shaped particles exhibited a basal- and prismatic-like crystal orientation, respectively, thereby supporting the results of several other works [2,8,9,10,11,12,14,15]. Accordingly, during the indentation experiments, special attention was paid to discerning the specific shape of the grain (i.e., crystal orientation) by means of AFM. The volume of the Co regions was too small to be tested by means of indentation. The indentation properties were extracted according to the ISO-14577 code [19]. A Poisson ratio of 0.2 was assumed for the WC phase for the post-processing of the indentation data [20].

The built-in Fit Gaussian Mixture Model to Data (fitgmdist) function was selected to post-process the measured nanohardness distribution of the 67 observations (hereinafter denoted as parent distribution) and to estimate the mean, co-variance matrix and the mixing proportion (the ratio between the number of elements in the parent distribution and that of the elements in each single/peak distribution) of each individual distribution that made up the mixture (hereinafter denoted as single/peak distribution). The built-in function of MATLAB was selected for the statistical study due its widespread use in academia and industrial research. The proposed algorithm [18] was found to be quite simple, robust and effective to achieve the present goal.

The *f*(*x*) mixture was composed of two normal probability density functions (*g_j_*(*x*)) with different means (*μ_j_*), co-variances (*Σ_j_*) and mixing proportions (*M_j_*):(1)$$f\left(x\right)=M_{1}g_{1}\left(x\right)+M_{2}g_{2}\left(x\right)$$
where *x* is the column vector containing *n* observations, and the *j*-th normal probability density function can be written as:(2)$$g_{j}\left(x\right)={\left[{\left(2\pi \right)}^{n}\left|\Sigma_{j}\right|\right]}^{-1/2}\;\exp\left\{-\frac{1}{2}{\left(x-\mu_{j}\right)}^{T}\Sigma_{j}^{-1}\left(x-\mu_{j}\right)\right\}$$
where *j* can be 1 or 2.

The employed expectation–maximisation algorithm was used to optimise the Gaussian mixture model likelihood by iterating over two-steps until convergence was reached. The a-posteriori probabilities ($w_{j}^{\left(i\right)}$) of the component memberships of the *j*-th single/peak distribution were first computed for each *i*-th element of *x*:(3)$$w_{j}^{\left(i\right)}=\frac{g_{j}\left(x\right)\;\Phi_{j}}{\sum_{h=1}^{2}g_{h}\left(x\right)\Phi_{h}}$$
where $\Phi_{j}\;$ is the average a-priori probability that an *i*-th element belongs to the *j*-th single/peak distribution:(4)$$\Phi_{j}=m^{-1}\overset{m}{\underset{i=1}{\sum }}w_{j}^{\left(i\right)}$$
where *m* is the number of observations that have a larger a-posterior probability than 50% of belonging to the *j*-th single/peak distribution. This was denoted as the expectation step. The means of the components, the covariance matrices and the mixing proportions were then estimated by applying the maximum likelihood procedure, assuming the component-membership a-posteriori probabilities as weights:(5)$$\mu_{j}=\frac{\sum_{i=1}^{m}w_{j}^{\left(i\right)}\;x^{\left(i\right)}}{\sum_{i=1}^{m}w_{j}^{\left(i\right)}}$$
(6)$$\Sigma_{j}=\frac{\sum_{i=1}^{m}w_{j}^{\left(i\right)}\;\left(x^{\left(i\right)}-\mu_{j}\right)\;{\left(x^{\left(i\right)}-\mu_{j}\right)}^{T}}{\sum_{i=1}^{m}w_{j}^{\left(i\right)}}$$
(7)$$M_{j}=\frac{m}{n}$$

The latter procedure is also denoted as the maximisation step. A trial-and-error scheme was designed, and it led to 1000 being chosen the as optimal iteration limit. With this iteration limit, the modelled results were found to be unaffected by the *μ_j_*, *Σ_j_* or *M_j_*, trial values, which were automatically selected randomly at the start. The elements belonging to each single/peak distribution were identified with the help of the *posterior* function. This built-in function computes the a-posteriori probability (Equation (3)) of each *i*-th element of the parent distribution to belong to each single/peak distribution. Each single/peak distribution is assumed to be composed of the elements that have a larger a-posteriori probability than 50% of belonging to it. The two single/peak indentation hardness distributions that were obtained were then analysed individually, by means of an independent Gaussian distribution function. The mean value of each single/peak distribution was equal to the mean values modelled by the fitgmdist function, and the standard deviation of each single/peak Gaussian distribution was equal to the root square of the corresponding element in the co-variance matrix of the parent distribution. Another analysis was then performed, to confirm that the actual parent distribution was a true two-single-peak distribution function, by forcing a three-single-peak distribution function. The result of this analysis was that the third nanohardness peak was in fact associated with a negligible mixing proportion of 0.02, which corresponded to only 1 element. This result confirmed that the parent distribution was of a bimodal type.

## 3. Results

The microstructure of the WC-Co composite, which shows a mixture of triangular and rectangular faceted WC particles, is displayed in Figure 1. The EBSD map in Figure 2 shows that the triangular particles are essentially characterised by {0001} (basal) planes, while the rectangular particles are composed of {1010} planes. The size distribution (400 ± 200 nm) of the WC particles shown in Figure 3 is based on three EBSD maps that cover an overall surface area of about 1000 μm^2^ (one map is shown in Figure 2, but the other two are not shown here). The EDS inspection in Figure 4, which is linked to the FESEM image in Figure 1, provides an analytical map and the concentration (wt%) of the relevant alloying elements that are present: a) on average (e.g., over the complete view field), b) in a WC particle (e.g., position A) and c) in the matrix (e.g., position B between WC particles).

The white 60-nm particles in Figure 1 are exogenous colloidal silica particles resulting from the previous chemomechanical polishing.

Figure 1 also shows the presence of voids among the WC particles (e.g., Position B). These originate during chemomechanical polishing, due to the preferential etching of the Co matrix by colloidal silica. More details on this phenomenon are given in the Discussion section.

Figure 5 shows the measured nanoindentation curves at the centre of both the triangular and rectangular WC particles (e.g., position A in Figure 1). The determined indentation hardness (*H_IT_*) and indentation modulus (*E_IT_*) distributions are displayed in Figure 6. At a first glance, the *H_IT_* parent distribution seems to exhibit a primary 22 GPa peak and a secondary smaller peak near 30 GPa. The MATLAB® fitgmdist built-in function was used and it identified a first single/peak at 22.76 GPa and a second single/peak at 29.61 GPa of 0.84 and 0.16 mixing proportions, respectively. Their elements in the co-variance matrix diagonal were 5.9 and 5.0 GPa, respectively. The bi-modal distribution and the single/peak distribution histograms are shown in Figure 6c, together with traces of the fitgmdist function and the respective Gaussian functions of each *H_IT_* single/peak. The two populations resulting from the fitgmdist function, based on the single/peak *H_IT_* distributions, are plotted in terms of *E_IT_* in Figure 6d. This figure shows that the indentation modulus shares similar bi-modal distribution features with *H_IT_*. A comparison of the indentation properties obtained in this study and those taken from the literature is reported in Table 1.

Figure 7 shows the indentation curves and the performed nanoimprints, as revealed by AFM force gradient forward scanning, over triangular and rectangular particles which, at the moment, we can only assume to be associated with basal-like oriented and prismatic-like oriented particles.

## 4. Discussion

It is well known that WC particles in sintered WC-Co composites have different crystallographic orientations that locally induce high anisotropic indentation properties [6,7,8,9,10,11,12,13,14,15]. It has in fact been confirmed that the hardness of a triangular face (basal plane) is qualitatively larger than that of a rectangular face (prismatic plane) [2,7,8,9,10,11,12,13,14,15]. Analogously, the indentation modulus increases from the prismatic to the basal crystal plane [2,8,11,12,15]. Cuadrado et al. [7] found that both basal and prismatic oriented particles exhibited a comparable indentation modulus, although the former was slightly smaller than the latter one. However, only one reference pointed out that the indentation hardness and indentation modulus over a prismatic plane were both larger than the basal ones [6].

The EBSD map (Figure 2) clearly shows that the triangular-shaped and rectangular-shaped particles are associated with basal-like and prismatic-like crystal planes, respectively, in agreement with refs. [2,8,9,10,11,12,14,15]. The present nanoindentation measurements reveal that the indentation properties of triangular particles, which so far have been assumed to consist of basal-like oriented grains, are larger than those of rectangular particles, which so far have been assumed to consist of prismatic-like oriented grains. Thus, as various authors [2,8,11,12,15] have shown that basal-like oriented grains are harder and stiffer than prismatic-like oriented grains, it is possible to conclude that the initial correspondence we assumed between the triangular and rectangular particles versus basal-like and prismatic-like oriented grains was sound.

However, a quantitative comparison of indentation properties from different literature sources is very difficult at present as a result of the multitude of factors that may influence common WC-Co composite materials of engineering interest, such as the chemical composition (WC/Co ratio and other additives), morphology, particle size and distribution, manufacturing process (with its inherent internal stresses and residual porosity), surface finish, indenter shape, indentation conditions (maximum load and loading rate), indentation size effect and even the procedure used to extract the properties.

Table 1 summarises the indentation results of a number of authors. It includes the indenter shape, the indentation method, the specification of the maximum load, the hardness over the main crystal planes and, when available, the indentation modulus. As can be seen, the hardness varies from 9.8 GPa, for a prismatic plane, to 57 GPa for a basal plane. The values of 30 and 23 GPa measured here over the basal and prismatic planes, respectively, compare well with 25 and 21 GPa [10], 34 and 25 GPa [12], and 28.9 and 21.9 GPa [14], all of which were obtained using a maximum load of 10 mN. A load as low as 4 mN gave comparable results (32.5 and 25.5 GPa, respectively) [15]. The measured indentation modulus was 670 and 578 GPa over the basal and prismatic planes, which are in good agreement with the 674 and 542 GPa values [8,11] obtained for an identical maximum load of 10 mN.

By taking advantage of the characteristic bi-modal distribution of nanohardness in WC particles, a statistical function was introduced that enables a distinction to be made between basal and prismatic planes [13,15]. Roa et al. [13,15] applied such a statistical analysis to a set of WC-Co composites, with 10 to 15 wt% Co and a particle size in the 1.1–2.3 μm range. Each composite grade of the whole set was nanoindented 1400 to 200,000 times on the basis of a specified regular grid. The determined nanohardness distribution displayed a characteristic three-peak distribution, that is, for the Co binder, the prismatic plane and the basal plane of the WC particles, respectively [13]. In a subsequent study, Roa et al. [15] found a five-peak distribution of the indentation hardness and indentation modulus for a WC- 11 wt% Co composite. The two additional peaks were derived from indentations at the boundary between the WC particle and Co or between WC particles with different orientations. In the present work, 67 nanoindentations were sufficient to detect the peaks of the basal and the prismatic planes of the WC particles. It is likely that the low Co content (10 wt%), together with the small WC particle size (400 nm) of our composite impeded the small Co regions from being nanoindented (even with the aid of an AFM). Consequently, the Co peak could not be determined, and no peaks associated with the boundary of the particles were found. Moreover, Figure 1 shows that the Co matrix had actually been replaced by voids between the WC particles, as it had been etched by colloidal silica during the chemomechanical polishing. Nevertheless, some large voids may also have been caused by the concomitant pulling out of some WC grains. A similar situation has been observed for a martensitic matrix in tool steels [16] which, because of the intrinsic different etching rate of its dispersed carbides, after chemomechanical polishing, presents a characteristic multi-level topological surface [16]. Figure 6 clearly indicates that the number of indentations over the prismatic planes is much larger than that over the basal ones, which is in fair agreement with refs. [13,15].

In principle, it may be expected that the mixing proportion of each single/peak distribution should be proportional to the fraction of either basal or prismatic oriented particles. However, this is true when indentations are performed over the whole surface, at the knots of a dense pre-specified regular grid. Here, the locations sampled for indentation were selected, using AFM, as being sufficiently flat, smooth and large to perform the test. Hence, the selected set of sampled locations does not represent the actual ratio between the basal- and the prismatic-oriented particles. However, it was reported that 0° to 17° tilted planes displayed a similar hardness to that of basal planes, and as a result the former are now denoted as “basal-like” [2]. Conversely, 37° to 90° tilted planes displayed a similar hardness to that of prismatic planes and, thus, are denoted as “prismatic-like” [2]. Assuming that the WC-Co has a random distribution of the WC particle orientations [6], it is evident that more particles exhibit a prismatic-like feature than a basal-like one, which is in qualitative agreement with our findings. Our EDS analysis confirmed that most of the Co binder is spread between the WC-particles. Alloy elements, such as Fe, V, Cr, Ni and Mn, are in solid solutions in both WC and Co (slightly richer in the binder), whereas Mo is only found in a small amount in WC particles [21].

## 5. Conclusions

The mechanical response obtained from nanoindentation tests on a full-density sintered WC-10Co composite rod, with specific reference to uniformly dispersed submicron WC particles, has been investigated as a function of the orientation of the WC crystal. The Co domains were too small to be indented, even over the nanoscale range. A new statistical tool has been employed to relate the indentation hardness and indentation modulus to the orientation of the WC crystal. A relatively small number of indentations, that is, fewer than 100, have proved to be sufficient to identify the indentation properties over each WC crystal plane. The results of the developed methodology confirm previous findings:

i) The observed two-peak distribution was associated with the anisotropic nature of the WC crystals;

ii) The {0001} basal plane exhibited grater indentation hardness and a larger indentation modulus than the {1010} prismatic plane.

## Figures and Tables

**Figure 1 materials-13-02137-f001:**
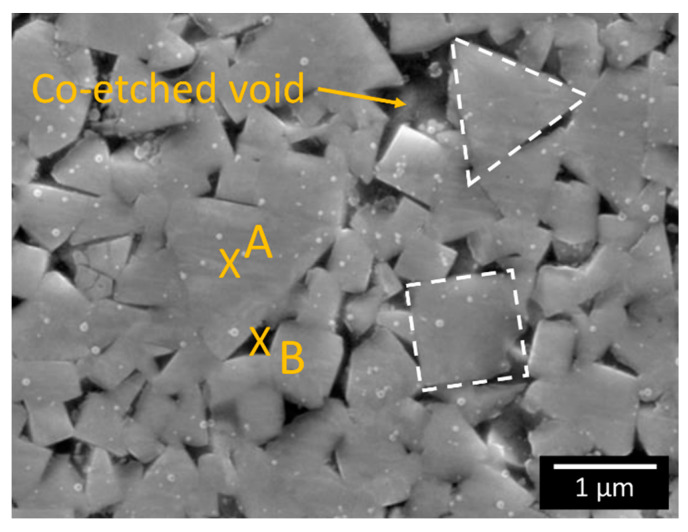
FESEM cross-sectional view of the base WC-Co microstructure after chemomechanical polishing. The dashed outlines show the triangular and rectangular grains. The white dots are residual SiO_2_ particles (60 nm size) left by the chemomechanical polishing. The arrow points to a Co-etched void that appeared after chemomechanical polishing.

**Figure 2 materials-13-02137-f002:**
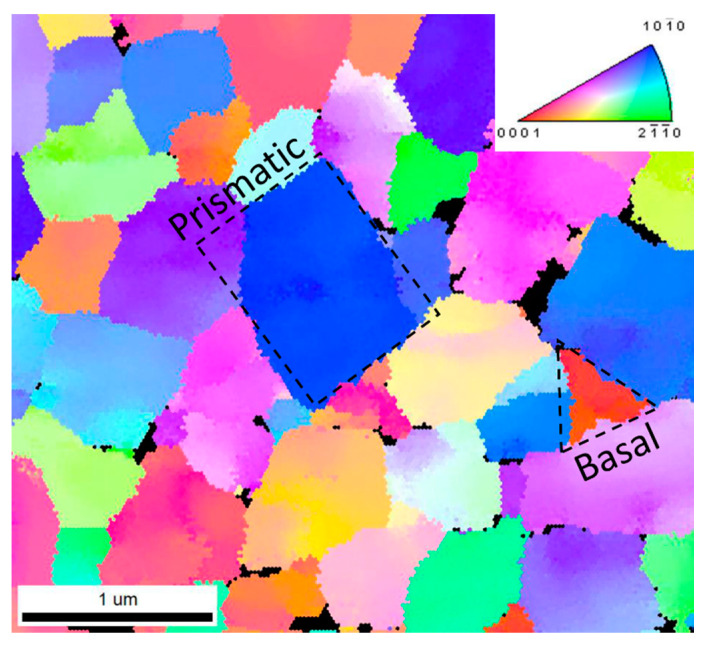
EBSD inverse pole figure (IPF) of the WC-Co microstructure after chemomechanical polishing.

**Figure 3 materials-13-02137-f003:**
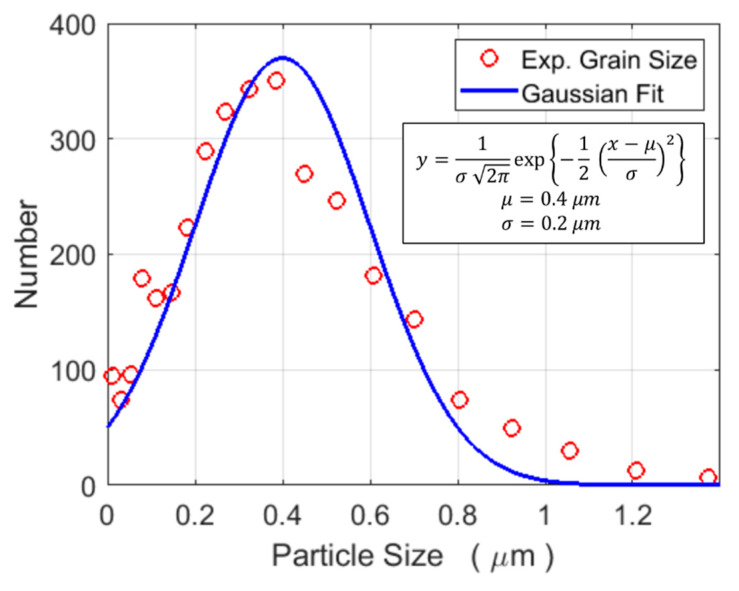
Experimental WC grain size distribution based on the EBSD inspection. The distribution contains information on three IPF maps (one shown in Figure 2, while the other two are not shown here) that cover an overall surface area of about 1000 μm^2^. In the Gaussian fit equation, *x* and *y* account for the “Number” and “Particle Size” axes, respectively, while *μ* and *σ* are the mean and standard deviation, respectively.

**Figure 4 materials-13-02137-f004:**
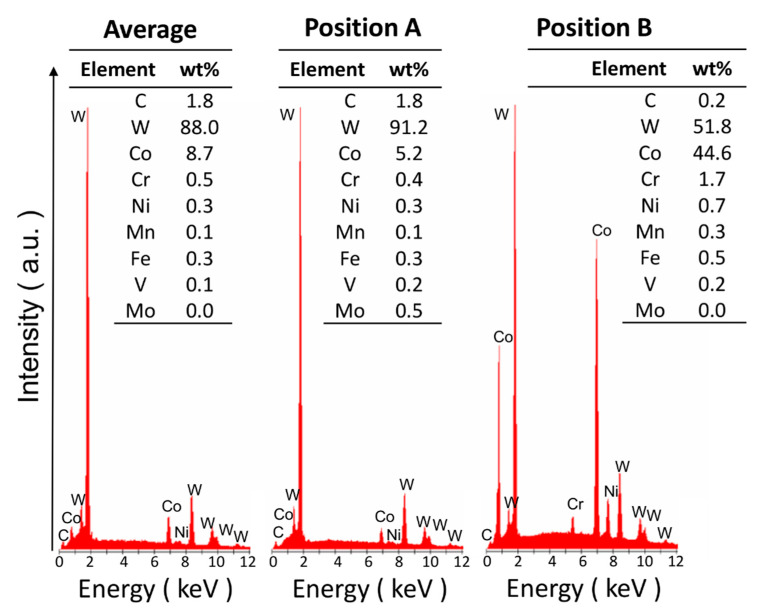
Comparison of the EDS spectra and semi-quantitative analysis for the various positions indicated in Figure 1. The analysis was repeated three times in each position and the results are listed here in terms of average weight percentages.

**Figure 5 materials-13-02137-f005:**
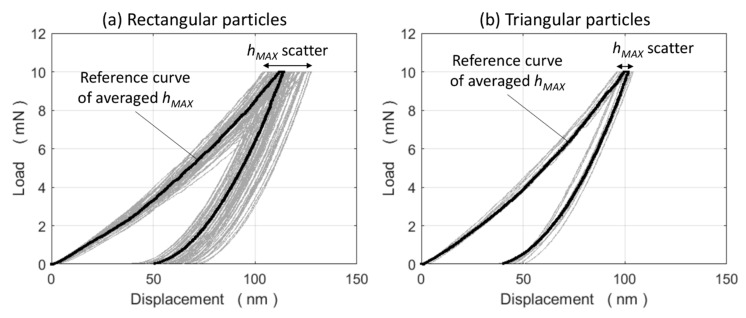
Nanoindentation curves of (**a**) rectangular and (**b**) triangular WC particles. An averaged maximum-indentation-depth (*h_MAX_*) reference curve is also shown.

**Figure 6 materials-13-02137-f006:**
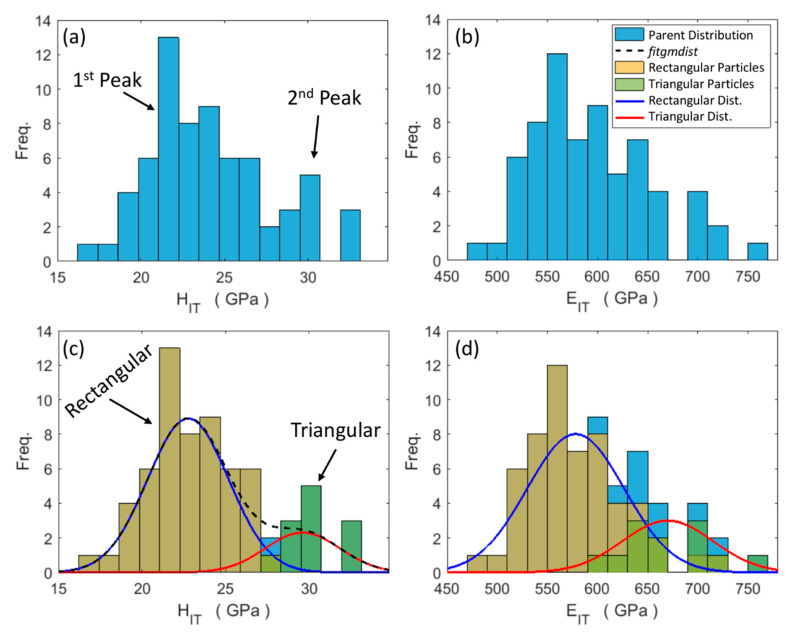
Parent distributions of both indentation hardness (**a**) and the indentation modulus (**b**). Superimposition of the parent distributions and the individual single/peak Gaussian distributions (blue and red lines), and inherent populations (orange and green bars), associated with rectangular and triangular particles, respectively, in the case of indentation hardness (**c**) and the indentation modulus (**d**). The black dashed line denotes the fitgmdist function trace.

**Figure 7 materials-13-02137-f007:**
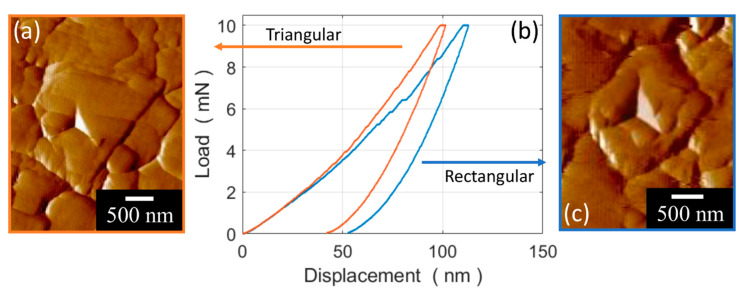
AFM force gradient forward scan of a typical nanoindentation imprint over (**a**) triangular and (**c**) rectangular WC particles, and (**b**) the inherent nanoindentation curves.

**Table 1 materials-13-02137-t001:** Indentation properties of the prismatic- and basal-like oriented particles, compared with literature data.

Test	Load	Hardness (GPa)		Indentation Modulus (GPa)		Ref.
		Basal	Prismatic	Basal	Prismatic	
Berkovich nIIT	10 mN	30 ± 2	23 ± 2	670 ± 45	578 ± 48	Present
Berkovich micro-IIT	250 mN	25.6 ± 0.2	17.2 ± 0.1	564 ± 26	532 ± 23	[7]
Vickers microhardness	200 mN	23.3	14.1	-	-	[9]
Berkovich nIIT	-	27	20	-	-	[13]
Berkovich nIIT	10 mN	40 ± 2	33 ± 2	674 ± 14	542 ± 34	[8,11]
Berkovich nIIT	5 mN	30 *	27 *	-	-	[10]
Berkovich nIIT	10 mN	25 *	21 *	-	-	[10]
Berkovich nIIT	50 mN	22 *	17 *	-	-	[10]
Berkovich nIIT	3 mN	57 ± 3	36 ± 2	-	-	[12]
Berkovich nIIT	3 mN	52.5 ± 2.5	33 ± 2	-	-	[12]
Berkovich nIIT	10 mN	43 ± 2	30 ± 2	-	-	[12]
Berkovich nIIT	10 mN	34 ± 2	25 ± 2	-	-	[12]
Berkovich nIIT	4 mN	32.5 ± 3.5	25.5 ± 5.0	650 ± 60	475 ± 80	[15]
Berkovich nIIT	10 mN	28.9 ± 0.1	21.9 ± 0.1	-	-	[14]
Berkovich nIIT	200 nm **	43.0 ± 0.8	28 ± 1	700	1050	[2]
Vickers macrohardness	10 N	22.2 ± 0.4	11.4 ± 0.5	-	-	[3]
Knoop microhardness	1 N	22 - 24.6	9.8–23.5	-	-	[5]
Vickers microhardness	1 N	21 ± 1	14 ± 1	-	-	[4]
Berkovich nIIT	300 μN	15–22 *	40–60 *	300–400 *	900–1200 *	[6]
Berkovich nIIT	900 μN	24 *	35–50 *	500 *	600–800 *	[6]

* Value taken from a picture that should only be considered as a reference value. ** Displacement control mode.

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
