# Peer review of "Nano-Indentation Properties of Tungsten Carbide-Cobalt Composites as a Function of Tungsten Carbide Crystal Orientation"

_materials, 2020, doi:10.3390/ma13092137_

Round 1

Reviewer 1 Report

The submitted manuscript entitled ‘Nano-Indentation Properties of WC Grains as a Function of Crystal Orientation in a WC-10Co Composite’ deals with the nano-indentation testing of sintered WC-Co samples with 10 wt% Co content. The Authors were able to distinguish the differently oriented WC particles based on their nanohardness and stiffness. The manuscript is interesting and logic, its main issue is the connection to the real-life applications. As in macroscale, the composite is isotropic, what is the reason behind the testing of particles individually? If any method could be suggested to how to orientate the single particles to have higher macroscale properties, that would be a really important scientific (and practical) application.

- ‘The built-in Fit Gaussian Mixture Model to Data (fitgmdist) function…’ – please insert the function as an equation. What is the reason behind the application of this function? Is there any reason to use this one instead of any other?

- In the case of fig 3 a curve fitting would be better than connecting the points.

- Please insert table 1 into fig 4 as a subfigure. Table 1 then could be omitted.

- this Reviewer suggests to plot all of the indentation curves by shifted origin. Or in another graph the average of the curves with its scatter band could be plotted.

- Line 207: ‘It19’ – ?.

Author Response

File attached.

Reviewer 2 Report

Overall, Scientific content is publishable.  However, the paper need general grammatical correction before publishing.   It would be great if authors can clarify the description of how the basal-like and prismatic-like particles were distinguished.  Please see below for the comments.

  • Page 2 line 52: hardness and the indentation modulus are higher?
  • Page 2 line 81: The WC powder means WC particle?
  • Page 2: Method: Authors described the WC-Co sample. Is it possible to provide more detail about the sample such as actual density
  • Page 3 line 93: Tip radius of 500nm. Is it 50 nm? Your indentation AFM image shows the tip radius is much smaller than 500 nm.
  • Page 3: line 97: Indentation size is depth dependent. It should be noted that the depending on the sample, and the tip shape/condition, apparent ISE may occur at different indentation depth.
  • Page 3 line 99: The projected area -> projected area function?
  • Page 3 line 101: authors have described 50 uNs-1 loading rate. This makes ~4 min indentation time.  How was thermal drift controlled?  With ~ 100 nm indentation depth,  thermal drift can significantly influence the measured properties. 
  • Page 4 line 140: It is hard to tell from SEM image that if Position B is a pull off region from polishing or Co matrix. Authors have stated it is a fully dense sample yet, the SEM image may incorrectly represent the sample.    It is possible smaller particles may have pulled off during polishing.  Also, it is unclear if the SEM image was taken with secondary electron detector or backscatter detector. If it is possible EDS mapping will help to show the Co matrix.
  • Page 4 line 160: could you clarify how basal-like and prismatic-like crystal planes were distinguished? Was every plane verified using EBSD or by shape?
  • Page 7 Figure 7: several load – displacement curve seems to show that there was either incorrect contact, or movement of sample (possibly particle) after indentation started. Also, due to the composite nature of the sample, the thicknesses of the indented particles are unknown.  Author should discuss about this point and try to remove those data points.
  • Page 9 line 207: typo – It19.

Author Response

File attached.

Reviewer 3 Report

Dear authors,
You raised the important question of the relationship of properties for hierarchical hard alloys from the crystal orientation of grains.
Studying presented results, the question arises. Why didn’t you pay due attention to the residual porosity of the samples and evaluate its effect for the model?

In addition, the results shown in table 1 should be rounded to tenths.
Good luck!

Author Response

File attached.

Round 2

Reviewer 1 Report

Thank you for all the changes and corrections, in the opinion of this Reviewer, the manuscript is now ready for publication.

However, the final decision belongs to the Editor of course.

Reviewer 3 Report

Dear authors,
You have made significant efforts to refine the manuscript. So the text looks structured and harmonious. In this form, the manuscript will be attractive to readers.
Good luck!